# Genome-Wide Identification of B-Box Gene Family and Expression Analysis Suggest Its Roles in Responses to Cercospora Leaf Spot in Sugar Beet (*Beta Vulgaris* L.)

**DOI:** 10.3390/genes14061248

**Published:** 2023-06-10

**Authors:** He Song, Guangzhou Ding, Chunlei Zhao, Yanli Li

**Affiliations:** 1College of Modern Agriculture and Ecological Environment, Heilongjiang University, Harbin 150080, China; 13104631380@163.com; 2Sugar Beet Engineering Research Center of Heilongjiang Province, Harbin 150080, China; 2002038@hlju.edu.cn (C.Z.); 1988011@hlju.edu.cn (Y.L.)

**Keywords:** B-box gene family, sugar beet genome, Cercospora leaf spot, bioinformatics

## Abstract

The B-box (BBX) protein, which is a zinc-finger protein containing one or two B-box domains, plays a crucial role in the growth and development of plants. Plant B-box genes are generally involved in morphogenesis, the growth of floral organs, and various life activities in response to stress. In this study, the sugar beet B-box genes (hereafter referred to as *BvBBXs*) were identified by searching the homologous sequences of the *Arabidopsis thaliana* B-box gene family. The gene structure, protein physicochemical properties, and phylogenetic analysis of these genes were systematically analyzed. In this study, 17 B-box gene family members were identified from the sugar beet genome. A B-box domain can be found in all sugar beet BBX proteins. *BvBBXs* encode 135 to 517 amino acids with a theoretical isoelectric point of 4.12 to 6.70. Chromosome localization studies revealed that *BvBBXs* were dispersed across nine sugar beet chromosomes except chromosomes 5 and 7. The sugar beet BBX gene family was divided into five subfamilies using phylogenetic analysis. The gene architectures of subfamily members on the same evolutionary tree branch are quite similar. Light, hormonal, and stress-related cis-acting elements can be found in the promoter region of *BvBBXs*. The *BvBBX* gene family was differently expressed in sugar beet following Cercospora leaf spot infection, according to RT-qPCR data. It is shown that the *BvBBX* gene family may influence how the plant reacts to a pathogen infection.

## 1. Introduction

Zinc-finger transcription factors are crucial for development, growth, and stress resistance [1]. Zinc-finger domains can interact with proteins, DNA, and RNA [2]. The B-box gene family is a subclass of zinc-finger proteins. It features a unique tertiary structure stabilized by zinc-ion binding and one or two conserved B-box domains [3]. This shows that the majority of the B-box domain in plants has been conserved. At the N-terminus, the B-box domain, which is made up of type 1 and type 2, has around 40 amino acids. The B-box domain is essential in the interaction of proteins and the control of transcription [4]. CCT (CONSTANS, CO-like, and TOC1) domains are also seen in several B-box proteins [5]. CCT domains were first discovered in *A. thaliana* CONSTANS, CO-like, and TOC1 proteins, which are essential for controlling flowering. At the C-terminus, the CCT domain has 42 to 43 amino acids.

The B-box protein has a crucial function in regulating plant growth and development. It is involved in seeding photomorphogenesis [6,7], photoperiodic regulation of flowering [8,9], shade avoidance [10], plant hormone signaling, and biotic and abiotic stress responses [11]. The yield of soybean grains can be increased by overexpressing *AtBBX32* in soybean, according to [12]. It has been documented that some species of plants can tolerate abiotic stress. *A. thaliana* can be more tolerant of abiotic stress when *MdBBX10* is overexpressed [13]. The expression of heat-responsive genes such as *DGD1*, *Hsp70*, *Hsp101*, and *APX2* is negatively regulated by *AtBBX18* in *A. thaliana*, which lowers seed germination and seedling survival after heat treatment [14]. The salt tolerance of *A. thaliana* can be improved by overexpressing *AtBBX24* (*STO*) [15]. *CmBBX24* performs two functions: delaying flowering and increasing plant resistance to cold and drought in chrysanthemums [16]. In grapevines, overexpression of *VvBBX32* enhanced cold tolerance [17]. A variety of abiotic stressors can cause sensitivity in rice B-box genes. Their transcriptional expression considerably changes in response to treatments for metal, salt, cold, drought, gibberellin (GA), salicylic acid (SA), and other stressors. Most rice B-box genes are highly expressed under abiotic stress [18]. B-box genes also play a role in biological stress. When the *IbBBX24* gene was overexpressed, it dramatically increased sweet potato resistance to Fusarium wilt [19]. The SA and ethylene (ETH) signaling pathways are mediated by the rice B-box gene *OsCOL9*, which improves blast resistance [20].

One of the two principal sugar crops grown worldwide is sugar beet. China mainly uses sugar beet to make sugar. It has a great position in agricultural production, because it can stabilize the development of the sugar industry and increase the income of the national economy. While growing and developing, sugar beet is subject to environmental stresses, including leaf spot disease, damping-off, and sugar beet root rot caused by *Rhizoctonia*. These stresses have a significant impact on sugar beet quality and yield. The fungus *Cercospora beticola* Sacc. is the source of Cercospora leaf spot (CLS) [21]. It is considered to be one of the most harmful sugar beet leaf diseases in the world. Cercospora leaf spot can reduce root yield by 10% to 20%, sugar content by one to two degrees, and stem and leaf loss by 40% to 70%. The roles of plant B-box proteins in flowering and mediating biotic and abiotic stresses are ideal agronomic traits in transgenic crops. In recent years, the B-box family members have been found in many plants, including rice [22], wheat [23], strawberry [24], tomato [25], sweet cherry [26], and *Chimonanthus praecox* [27]. The sugar beet B-box gene family has not undergone an extensive study. The sugar beet genome is used in this study’s analysis of the B-box gene family as a reference for future research on the genes’ functions. The obtained results hold great significance for the future breeding and improvement of new sugar beet varieties.

## 2. Materials and Methods

### 2.1. Identification and Physicochemical Properties Analysis of BvBBXs

We downloaded sugar beet DNA sequences, protein sequences, and annotation files from the Ensembl Plants database (https://plants.ensembl.org/index.html, accessed on 6 January 2022) [28]. Two methods were used to distinguish the B-box gene family members in sugar beet. Method 1: the hidden Markov model file for the zf-B_box (PF00643) was acquired from the Pfam database (https://www.ebi.ac.uk/interpro/entry/pfam/#table, accessed on 6 January 2022), and the sugar beet protein sequence was searched using HMMER software. We then obtained the highest-matching protein sequence in sugar beet [29]. Method 2: the B-box family members in *A. thaliana* were used as information probes, and the homologous sequences in the sugar beet protein database were searched with blastp (the minimum E value was e^−3^) [30]. A total of 31 potential sequences were discovered after combining the findings of the two approaches mentioned above. The protein sequences’ conserved domains were examined in the Pfam database and the online tool, CDD, offered by the NCBI (https://www.ncbi.nlm.nih.gov/Structure/cdd/wrpsb.cgi, accessed on 7 January 2022), and any sequences lacking the B-box conserved domain were eliminated [31].

The Expasy website (http://expasy.org/, accessed on 15 January 2022) was used to examine the physical and chemical properties of the sugar beet B-box amino acid sequence, including isoelectric point and molecular weight predictions [32]. The online tool Cell-PLoc 2.0 (http://www.csbio.sjtu.edu.cn/bioinf/Cell-PLoc-2/, accessed on 10 May 2022) was used to determine the subcellular position by inputting the amino acid sequence of the sugar beet B-box gene family.

### 2.2. Chromosome Localization Analysis of BvBBXs

The sugar beet gene information GFF3 file was used to extract the *BvBBXs’* chromosomal position information. The TBtools Gene Structure Shower tool was used to view the chromosomal location data of *BvBBXs.*

### 2.3. Domain Analysis of BvBBXs

Through the domain-initiation sites provided in the CDD, we can sort out the domain information of the B-box gene family and use WebLogo (https://weblogo.berkeley.edu/logo.cgi, accessed on 3 February 2022) to visualize this information in order to better understand and analyze the domain information [33]. The domains of sugar beet BBX proteins were visualized using TBtools visualize Domain Pattern tool.

### 2.4. Analysis of Gene Structures and Conserved Motifs of BvBBx Genes

The sugar beet gene information GFF3 file was used to extract the gene structure information for *BvBBXs*, and the TBtools Gene Structure Shower tool was used to see the structure information (exons, introns) for *BvBBXs* [34]. The conserved domains were estimated using the MEME online tool (https://meme-suite.org/meme/doc/meme.html, accessed on 5 February 2022). Ten motifs were predicted. We visualized the motif pattern of sugar beet BBX proteins with the Visualize Motif Pattern tool of TBtools.

### 2.5. Alignment and Phylogenetic Analysis BvBBXs

Rice and *A. thaliana* whole genome sequence files as well as annotation files were downloaded from NCBI (https://www.ncbi.nlm.nih.gov/, accessed on 15 February 2022). *AtBBXs* in *A. thaliana* are from the study by Khanna [2]. *OsBBXs* in rice were studied by Jianyan Huang [22]. The MEGA11 program was used to create the phylogenetic tree. First, the B-box gene sequences of sugar beet, *A. thaliana* and rice were aligned using ClustalX software. To create a phylogenetic tree using the neighbor-joining algorithm and to obtain more precise findings, we set up a thousand bootstrap repeats [35,36]. By using the subfamily classification outcomes for the B-box gene family in *A. thaliana*, rice, and wheat [2,22,37], the subfamily classification of the B-box gene family in sugar beet was carried out. The online software iTOL (https://itol.embl.de/, accessed on 1 April 2022) was used to beautify the phylogenetic tree.

### 2.6. Analysis of Cis-Acting Elements of BvBBXs’ Promoters

The promoters’ sequences were considered to be 2000 bp upstream of the first CDS of the *BvBBXs* and was gained from the genome file and genome annotation file with TBtools. The PlantCARE database (https://bioinformatics.psb.ugent.be/webtools/plantcare/, accessed on 17 February 2022) was used to examine the type, quantity, and function of cis-acting elements [38]. We used TBtools’ Simple Bio Sequence Viewer to visualize the information.

### 2.7. Analisis of BvBBx Genes’ Potential Role in Leaf Spot Disease Resistance

#### 2.7.1. Infection Experiment

We used two sugar beet varieties, KWS5145 (highly susceptible to Cercospora leaf spot) and F85621 (highly resistant to Cercospora leaf spot). We divided them into three groups for the experiment, namely, the control group, the early-infection test group, and the disease test group. The fungi were introduced using the spray method when the seedling age was three pairs of true leaves. The early-infection test group and the disease test group were evenly sprayed with the spore suspension, and the control group was sprayed with the same amount of distilled water. The plants were raised in a light culture environment with a humidity level of 90% and a temperature of 26 °C.

#### 2.7.2. Sampling Treatment

The control group (no *C. beticola* infection): 1 g of the leaves in the same growth status were individually excised from three plants (that is, 3 samples in each group) for each sugar beet variety. The highly susceptible variety, KWS5145, was numbered HS_CK, and the highly resistant variety, F85621, was numbered HR_CK. Each sample was stored in a separate centrifuge tube at −80 °C. 

The early-infection test group: *C. beticola* began to colonize at 36 h after infection; 1 g of the leaves in the same growth status were individually excised from three plants (that is, 3 samples in each group) for each sugar beet variety. The sample of the highly susceptible variety, KWS5145, was named HS_infected, and the sample of the highly resistant variety, F85621, was named HR_infected. All the samples were saved at −80 °C. 

The disease test group: The symptoms began to appear at 96 h after infection; 1 g of the leaves in the same growth status were individually excised from three plants (that is, 3 samples in each group) for each sugar beet variety. They were numbered in separate centrifuge tubes. The highly susceptible variety, KWS5145, was numbered HS_disease. The samples of the highly resistant variety, F85621, were numbered HR_disease. All the samples were saved at −80 °C. 

#### 2.7.3. Quantitative Real-Time PCR (RT-qPCR) Analysis

Liquid nitrogen was used to homogenize plant materials that had been added to a mortar. The amount of powder that was put into a microtube was about 100 mg. According to methods in Kobayashi et al. (2010) [39], total RNA isolation and purification were carried out. Additionally, using the PrimeScript RT Reagent Kit (Takara, Dalian, China) in accordance with the manufacturer’s instructions, the RNA was reverse transcribed into cDNA. The PCR reaction solution was prepared according to Table 1. A three-step PCR amplification reaction was performed. PCR conditions were as follows: Predenaturation was performed at 95 °C for 30 s. Then, 40 cycles of PCR amplification were performed at 95 °C for 5 s and 60 °C for 30 s. After the amplification cycle, it was cooled to 60 °C, and then heated to 95 °C to denature the DNA product. The specific primers used for quantitative real-time PCR analysis are shown in Table 2. Gene expression was generally measured in three independent biological replicates, and at least two technical replicates were used for each of the biological replicates. After the reaction, the real-time qPCR amplification and melting curves were verified.

### 2.8. Data Analysis

The statistical analysis was performed using the IBM SPSS Statistics 26.0 software package, and Duncan’s multiple range tests were utilized to determine the significance of differences. Statistical significance was defined as a *p*-value of 0.05 or less, and extreme significance as a *p*-value of 0.01.

## 3. Results

### 3.1. Identification and Physicochemical Properties of B-Box Gene Family in Sugar Beet Genome

According to the hidden Markov model (HMM)-based profile (B-box-type zinc-finger domain, PF00643), the genome-wide annotated proteins were screened and verified using Pfam databases. The 17 members of the sugar beet B-box gene family were discovered. They were given the names *BvBBX1-17* based on where they were located on the chromosome.

The genes’ physicochemical properties information listed in Table 3 includes the gene symbol, accession number, number of amino acid residues, isoelectric point (pI), and molecular weight (MW) of the protein. The predicted outcomes revealed that all sugar beet BBX proteins have isoelectric points lower than 7, indicating that all sugar beet BBX proteins were weakly acidic proteins. The molecular weight of *BvBBXs* ranged from 14.74 kDa (*BvBBX17*) to 57.63 kDa (*BvBBX08*). The number of amino acid residues of sugar beet BBX proteins ranged from 135 to 517 amino acids. There is a great disparity. The results of the subcellular localization prediction using online software Cell-PLoc 2.0 showed that 17 *BvBBXs* were localized in the nucleus.

### 3.2. Chromosome Localization Analysis of B-Box Gene Family in Sugar Beet

The sugar beet B-box gene family can be found on chromosomes based on the physical position information. The 17 *BvBBXs* were found to be irregularly dispersed throughout seven chromosomes (Figure 1). The chromosome locations of *BvBBX03*, *BvBBX06*, *BvBBX10*, *BvBBX11*, *BvBBX16*, and *BvBBX17* are unknown, and the remaining 11 genes are located on seven of the nine chromosomes of sugar beet. There is one member on chromosomes 1, 2, 3, 4 and 8 respectively. There are three members on chromosome 6 and 9 respectively. *BvBBX07*, *BvBBX08,* and *BvBBX09* were found on chromosome 6. *BvBBX13*, *BvBBX14,* and *BvBBX15* were found on chromosome 9. No *BvBBX* gene was found on chromosome 5 and chromosome 7.

### 3.3. Domain Analysis of Sugar Beet B-Box Family Members

Six of the 17 *BvBBXs* (*BvBBX10*, *BvBBX08*, *BvBBX13*, *BvBBX03*, *BvBBX12*, and *BvBBX02*) have both a conserved CCT domain and two B-box domains. Two B-box domains, but no CCT domain, are present in five members (*BvBBX15*, *BvBBX01*, *BvBBX05*, *BvBBX04*, and *BvBBX14*). There is just one B-box domain in the four members (*BvBBX07*, *BvBBX17*, *BvBBX06*, and *BvBBX11*). The remaining two members protein (*BvBBX09* and *BvBBX16*) contain one B-box domain and one CCT domain (Figure 2).

Sugar beet BBX proteins have highly conserved B-box and CCT domains, according to the protein sequence alignment. The conserved sequences in the B-box1 and B-box2 domains are similar (Figure 3). The conservation of all amino acid residues in Figure 3 is represented by the height of each letter. The results show 38 conserved motifs in B-box1 and B-box2. The common sequence of B-box1 comprises C-X2-C-X8-C-X2-D-X1-A-X1-L-C-X2-C-D-X3-H-X8-H. The common sequence of B-box2 comprises C-X2-X8-X7-C-X2-C-X4-H-X8-H. B-box1 is more conservative. It also contains the conserved residues of B-box2. The common sequence of the CCT domain is R-X5-R-Y-X1-E-K-X5-R-X3-K-X6-R-K-X2-A-X6-K-X1-R-X2-K.

### 3.4. Analysis of Gene Structure and Protein Conserved Motifs of Sugar Beet B-Box Family Members

The diversity of gene structure is what drives the evolution of multigene families. The structure of the *BvBBXs* was analyzed (Figure 4). The number of exons was one to five, and the number of introns was zero to four. In general, members of each subfamily have similar genetic structures.

To deepen our understanding of the diversity and conservatism of sugar beet BBX protein structure, the MEME online program was used to predict the conserved motifs of the BBX proteins. In this study, 17 sugar beet BBX proteins contain a total of 10 significant conserved motifs, Motif1 to Motif10 (Figure 4). Motif1 and Motif4 were found in all proteins. About 60% of proteins contain Motif3. Motif2 is present in subfamily I, II, and III. In subfamily IV, Motif3 is present. The motif composition of other members is roughly the same expect *BvBBX07*. In subfamily V, all genes contain Motif1, Motif4, and Motif10 except *BvBBX11.* It can be seen that the composition of conserved motifs is high in proteins with close evolutionary relationships.

### 3.5. Classification and Phylogenetic Analysis of Family Members

A phylogenetic tree was built for 17 sugar beet BBX protein sequences in order to thoroughly examine the evolutionary relationship and functional variations of members. According to the grouping research and analysis of Khanna et al., the B-box family members in sugar beet could be separated into five subfamilies (Figure 5). Each of the five subfamilies has three, five, one, five, and three members, respectively. The members of subfamily I include *BvBBX02*, *BvBBX03*, and *BvBBX12*, which contain two B-box domains and one CCT domain. Subfamily II includes *BvBBX07*, *BvBBX13*, *BvBBX10*, *BvBBX08*, and *BvBBX09*. Subfamily II members contain two B-box domains and one CCT domain, except for *BvBBX07*. Subfamily III has only one member (*BvBBX16*), which contains only one B-box domain and one CCT domain. Subfamily IV has five members, including *BvBBX14*, *BvBBX15*, *BvBBX01*, *BvBBX04*, and *BvBBX05*. They lack the CCT domain and only have two B-box domains. Subfamily V has three members (*BvBBX06*, *BvBBX13*, and *BvBBX11*), each containing only one B-box domain and no CCT domain. The subfamily classification is similar to that of the *A. thaliana* B-box gene family except for subfamily II. It is important to note that *BvBBX07* is closer to subfamily II in phylogeny even though, according to its structural classification, it belongs to *A. thaliana* subfamily V.

### 3.6. Cis-Acting Element Analysis

We examined the 2000 bp upstream sequence of the CDS of the *BvBBXs* (as a promoter) to learn more about the function of the sugar beet B-box gene family. A total of 85 cis-acting elements were predicted in *BvBBXs*. There are 27 cis-acting elements appearing in at least three members (Figure 6). A total of 12 light-response elements, six hormone-response elements, five defense-response elements, and two developmental elements are expressed in *BvBBXs*, excluding the main promoter elements, the CAAT-box and TATA-box (Table 4). Among the 17 *BvBBXs* promoters, the highest number of cis-acting elements were light-response elements, such as Box-4, G-box, the TCT-motif, I-box, CATA-box, AE-box, TCCC-box, MRE, ACE, the ATCT motif, CCAAT-box, and the GT-1 motif. The hormone-response elements include various elements that participate in the response to abscisic acid (ABS), methyl jasmonate (MeJA), salicylic acid (SA), gibberellin (GA), and auxin (IAA). Examples of such elements include the TGACG motif, TCA element, ABRE, CGTCA motif, TGA element, and P-box. Among the defense-response elements, MBS is related to drought induction, LTR is involved in the low-temperature response, ARE is related to anaerobic induction, the TC-rich repeat is a defense and pressure cis-acting element, and the WUN motif is related to wound induction. There are also CAT-boxes related to meristem expression and O_2_-site developmental cis-acting elements involved in the regulation of zein metabolism. According to our research, the *BvBBXs’* promoter region is crucial for the hormone, stress, and light responses. 

### 3.7. Expression Analysis of BvBBXs Gene in Highly Resistant and Susceptible Varieties

To analyze the potential role of the *BvBBX* genes in Cercospora leaf spot resistance, we performed RT-qPCR to identify the expression patterns of the *BvBBXs* in high-susceptible and high-resistant varieties after 36 and 96 h of *C. beticola* Sacc. infection (Figure 7).

The 15 *BvBBX* genes, with the exception of *BvBBX15* and *BvBBX01*, were expressed to various degrees in the leaf tissues of the two varieties. The expression of *BvBBX4* was the highest, and the expression of *BvBBX08* was the lowest. Three genes *BvBBX10*, *BvBBX07*, and *BvBBX06* were significantly upregulated after infection with Cercospora leaf spot. Only *BvBBX05* was significantly downregulated. Another gene, *BvBBX02*, was upregulated at 36 h after infection and downregulated at 96 h after infection. *BvBBX12* and *BvBBX11* expression levels were upregulated and then downregulated. The expression levels of the two genes *BvBBX08* and *BvBBX09* remained basically unchanged after infection. 

Four genes were differentially expressed in highly susceptible and highly resistant varieties (Figure 8). At 36 h after infection with Cercospora leaf spot, the expression of *BvBBX04* was noticeably higher in the highly susceptible variety than in the highly resistant kind. When infected with *C. beticola* for 96 h, the expression level of *BvBBX03* in the highly susceptible variety was noticeably lower than that in the highly resistant variety. At 36 h and 96 h after infection, the expression level of *BvBBX16* in the highly resistant variety was significantly higher than that in the highly susceptible variety. After 36 h and 96 h of infection, the expression of *BvBBX17* in the highly resistant variety was significantly lower than that in the highly susceptible variety. Therefore, the differential expression of *BvBBX04*, *BvBBX03*, *BvBBX16*, and *BvBBX17* could be related to the disease resistance of sugar beet.

## 4. Discussion

In this study, the sequences of the *BvBBX* genes were analyzed primarily using bioinformatics techniques. The structure and function of the sugar beet B-box gene family were predicted using phylogenetic analysis, gene structure analysis, and cis-acting element prediction. The results of the study indicate the presence of 17 *BvBBXs* in the sugar beet genome. Genes were unevenly distributed on seven chromosomes of the sugar beet genome, but no *BvBBX* gene was found on chromosomes 5 and 7. These sugar beet BBX proteins are distinguished by one or two B-box domains at the N-terminus and, occasionally, a CCT domain at the C-terminus. This is consistent with the characteristics of the B-box gene family. A total of 32 *A. thaliana* B-box proteins were found in *A. thaliana* [2]. The BBX protein has also been reported in other plants. The BBX gene family has been identified in rice [22], pear [40], apple [41], tomato [25], cherry [26], etc. The number of BBX gene families varies with plant species. BBX family members have been identified in several plant species, including 30 in rice, 25 in pear, 64 in apple, 29 in tomato, and 15 in cherry. These differences could be due to differences in genome size and complexity between these species. Species-specific duplication or deletion during evolution could have led to differences in the numbers of the BBX family of transcription factors across various species. According to the similarity of amino acid sequences and different combinations of conserved domains, sugar beet BBX proteins can be further divided into five families. The classification of the sugar beet BBX gene family is similar to that of *A. thaliana*. Five subfamilies of the B-box gene family were created in *A. thaliana*. All members of subfamily I and subfamily II contained two B-box and CCT domains. Subfamily III had one B-box and one CCT domain. Subfamily IV and subfamily V had two and one B-box domains, respectively [2]. *BvBBX7*, however, belongs to subfamily II and has a B-box domain but no CCT domain. Only the classification of subfamily II is different from that of *A. thaliana*. Similar inconsistencies were found in other species. *OsBBX25* and *OsBBX27*, for instance, are members of subfamily I in rice. *ZmBBX7* in maize belongs to subfamily II. Nevertheless, none of them have a B-box domain [5]. Despite possessing two B-box domains, *SlBBX9, SlBBX11,* and *SlBBX12* in tomato, as well as *OsBBX7* and *OsBBX19* in rice, are categorized under subfamily III of the BBX family [42]. It could be that the lack of repetition in evolution leads to the difference in grouping.

The evolution of gene families is significantly influenced by the diversity of gene structure. This study revealed that the subfamily of B-box genes in sugar beet displays a high degree of similarity in both gene structure and functional characteristics, characterized by a reduced number of introns and an abundance of conserved sequences. This is consistent with the gene structure of the B-box gene family in other plants. The B-box gene family displayed a significant degree of sequence similarity, according to multiple sequence alignment. The B-box1 and B-box2 domains are consistent with the characteristics of *A. thaliana*. B-box1 is more conserved than B-box2, and the CCT domain is also highly conserved. We speculate that B-box1 and B-box2 sequences could be derived from fragment duplication and internal deletion events. Therefore, we believe that members of BBX have similar gene structures and similar functional characteristics based on evolutionary relationships.

The kind and quantity of cis-acting elements in the promoter region can result in variable expression of genes, and these elements influence gene transcription. It was discovered that the sugar beet BBX gene family’s promoter regions comprised various numbers and kinds of cis-acting elements. Here, we discovered that the proportion of light-responsive cis-elements in the promoters of the 17 *BvBBXs* was the highest. This proves that some BBX proteins do play a central role in various photoregulatory physiological processes in plants. Apart from the light-response elements such as Box4, G-box, I-box, the GATA motif, and ATCT motif, numerous elements associated with hormone and stress response were detected in the promoter region of multiple BvBBXs, including the TCA element, ABRE, CGTCA motif, GT-1 motif, and others. The examination of cis-acting elements revealed that *BvBBX* genes were crucial in the control of sugar beet development under hormonal and stressful conditions, in addition to their involvement in the regulation of light signals. Some cis elements linked to light response, stress response, and hormone response were discovered in the promoters of the majority of BBXs in rice, tomato, jasmine, cherry, and other plants. In *Chimonanthus praecox*, *CpBBX19* contains a defense- and stress-response element (TC-rich), a salicylic acid response element (TCA element), four drought-induced MYB binding sites (MBSs), an abscisic acid response element (ABRE), two MeJA response elements (the TGACG motif and CGTCA motif), and the expression of *CpBBX19* is induced by abiotic stress (drought, salt, cold, and heat) and exogenous hormones (ABA and MeJA) [27]. This suggests that *BvBBX* genes may be involved in the regulation of hormones, abiotic stress, and the light response. These cis-acting regions can activate or repress genes in response to particular stressors. Numerous studies have shown that abiotic stressors and other hormones can cause the expression of the BBX genes. Low-temperature stress induced the expression of *SlBBX7*, *SlBBX9,* and *SlBBX20* in tomatoes [25]. The expression of some *GhBBX* genes in upland cotton was different after abscisic acid (ABA) treatment [43]. Treatment with ABA, GA, and BR led to the expression of *PavBBX6* and *PavBBX9* in sweet cherry [26]. By treating apples with high salt, low temperature, and exogenous ABA, 12 *MdBBX* genes were induced [41]. In this study, the B-box family members of highly resistant and highly susceptible beet varieties were induced and expressed to different degrees after infection with Cercospora leaf spot. Three genes, including *BvBBX6*, *BvBBX7*, and *BvBBX10*, were significantly upregulated after infection with *C. beticola*. One gene, *BvBBX05*, was significantly downregulated. This indicated that *BvBBX* genes may play a role in the response to pathogen attack. Four differentially expressed genes were found between highly susceptible and highly resistant varieties, which may be related to the resistance of sugar beet varieties to Cercospora leaf spot. By controlling the expression of genes associated with plant hormones, several transcription factors enhance the capacity of plants to defend themselves. *IbBBX24* strengthens sweet potato’s resistance to Fusarium wilt by encouraging the jasmonic acid pathway [19]. Therefore, these four genes may regulate the resistance to Cercospora leaf spot through a hormone pathway. Numerous transcription factors, including WRKY, NAC, and bZIP, are crucial in the interactions between hosts and pathogens. However, it is still unknown how B-box transcription factors perform in terms of disease defense. Instead, it is known that they play a significant role in the control of light response, hormone response, and abiotic stress. 

The significance of BBX proteins in other plants is still yet to be fully understood, despite great progress being made in understanding the activities of several BBX proteins in various developmental processes in *A. thaliana*. In the upcoming years, it will be crucial to comprehend the molecular workings of each BBX protein. In conclusion, this research paved the way for future research on the sugar beet BBX genes, which is crucial for creating novel sugar beet varieties. 

## 5. Conclusions

In this study, the sugar beet genome was examined, and 17 members of the sugar beet BBX family, which may be further subdivided into five subfamilies, were screened. Similar to the subfamily classification of *AtBBXs*, *BvBBX7* is closer to subfamily II in phylogeny, but has only one B-box domain and no CCT domain. This is different from the domains contained in the subfamilies of phylogenetics in *A*. *thaliana*. Except for chromosomes 5 and 7, chromosome localization studies revealed that *BvBBXs* were dispersed on seven chromosomes. *BvBBXs* had one to five exons and zero to four introns. Members in the same subfamily have similarities in gene structure. The promoter region of *BvBBXs* contained numerous cis-acting elements involved in the light, hormone, and stress responses, and the expression of the sugar beet B-box gene family changed following Cercospora leaf spot infection. Three genes, *BvBBX6*, *BvBBX7*, and *BvBBX10* were significantly upregulated after infection, and one gene, *BvBBX5*, was significantly downregulated. The findings suggested that the *BvBBX* genes may be involved in the defense against pathogen attack, and it has been hypothesized that they may control the stress response process to Cercospora leaf spot infection. The sugar beet BBX family’s genome-wide investigation will establish the groundwork for future research into the role of BBX genes in sugar beet.

## Figures and Tables

**Figure 1 genes-14-01248-f001:**
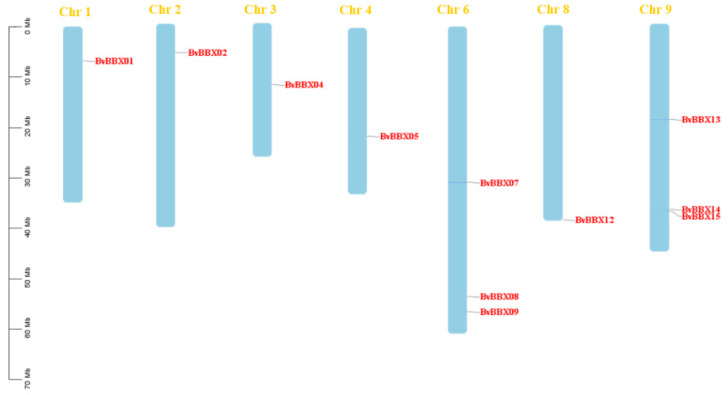
Chromosomal localization of the sugar beet B-box gene family.

**Figure 2 genes-14-01248-f002:**
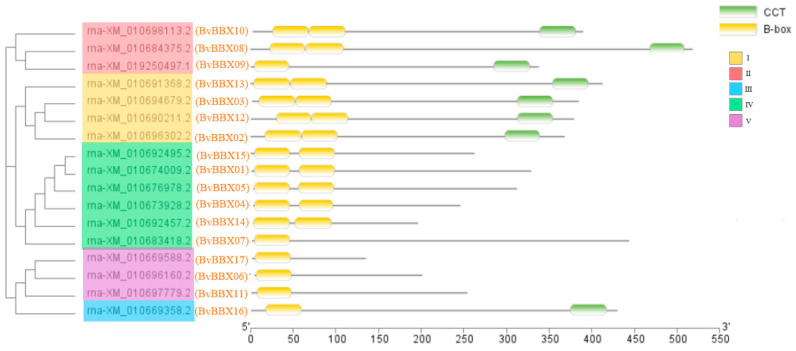
Domains of the sugar beet B-box gene family.

**Figure 3 genes-14-01248-f003:**
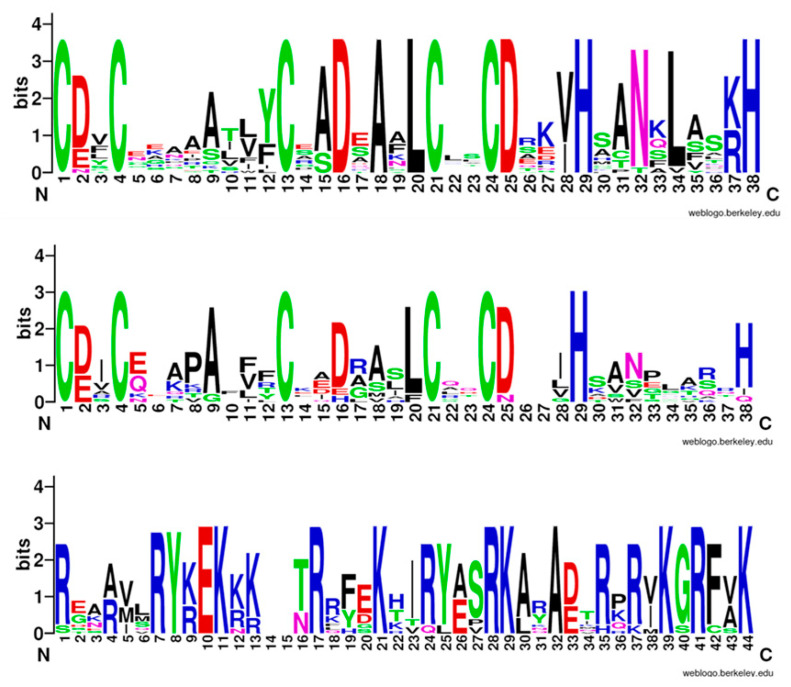
Conserved domains of B-box gene family in sugar beet.

**Figure 4 genes-14-01248-f004:**
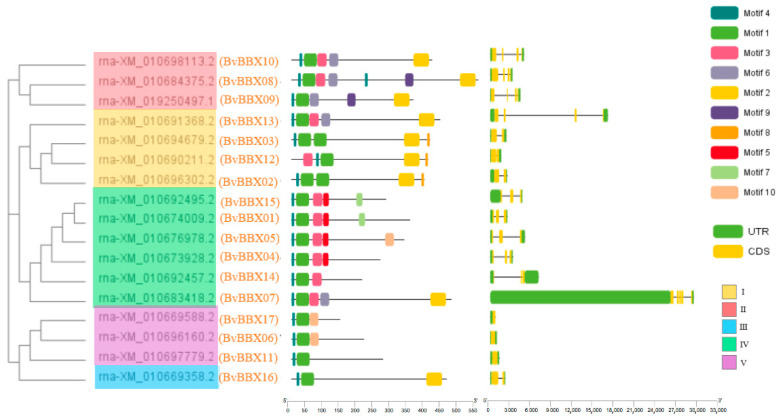
Motif analysis and gene structure of sugar beet B-box gene family.

**Figure 5 genes-14-01248-f005:**
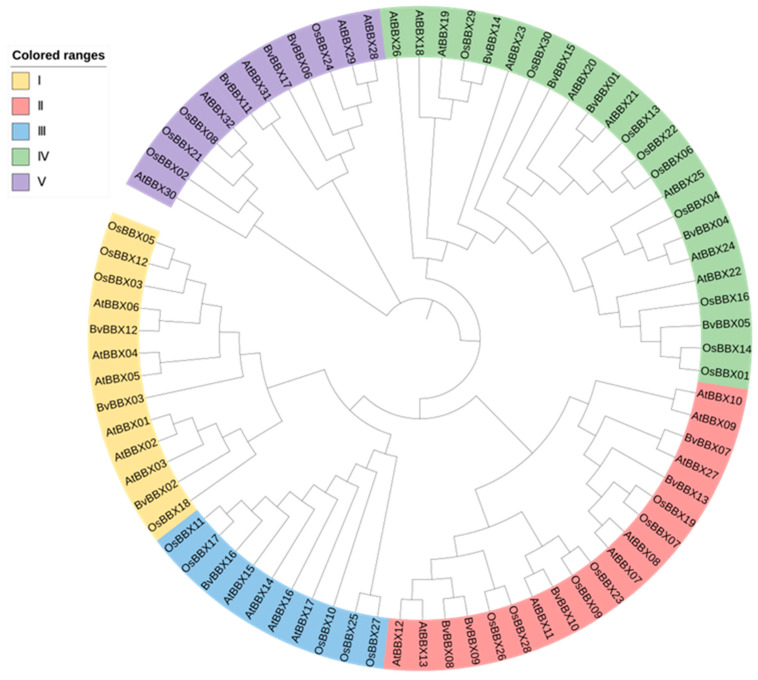
Phylogenetic tree of the sugar beet, rice, and *A. thaliana* B-box gene family.

**Figure 6 genes-14-01248-f006:**
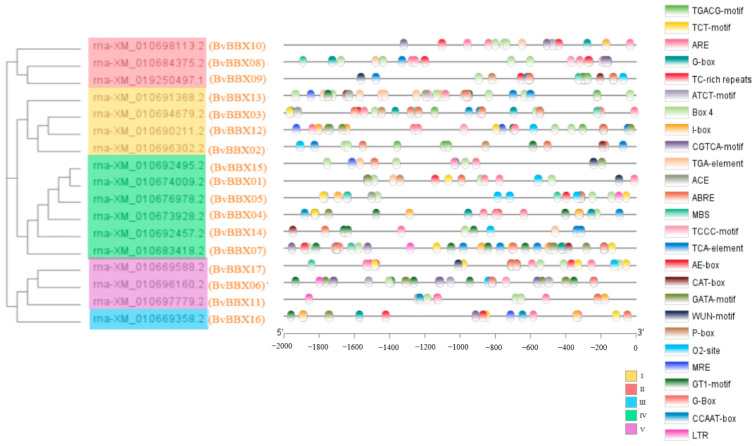
Cis-acting element analysis of B-box gene family in sugar beet.

**Figure 7 genes-14-01248-f007:**
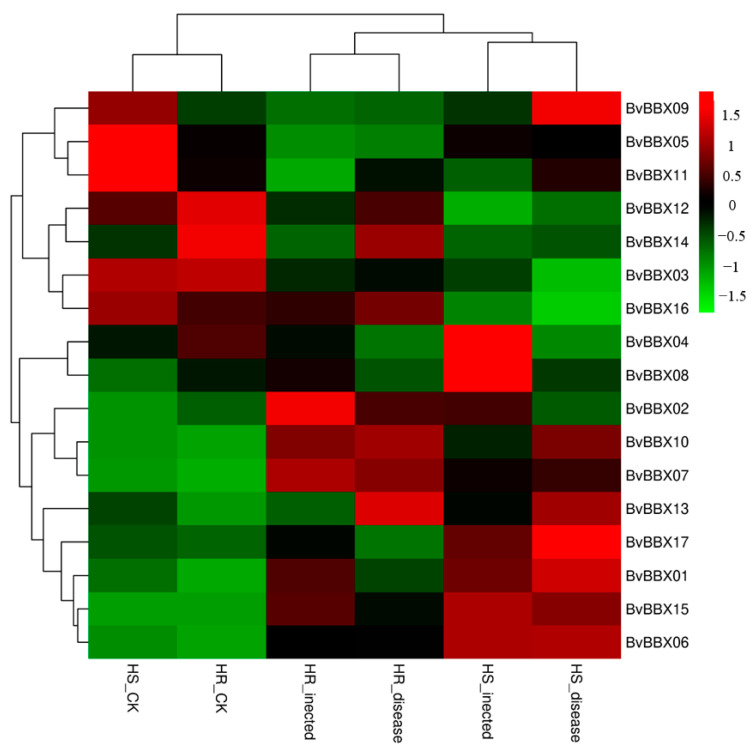
The expression patterns of *BvBBXs* in highly resistant and highly susceptible varieties of sugar beet. Note. The x-axis shows the sample, the y-axis is the relative expression level of each gene in leaf tissue. “HS” refers to the highly susceptible variety; “HR” refers to the highly resistant variety; “infected” represents the infection of sugar beet brown spot for 36 h; “disease” represents the infection of sugar beet brown spot for 96 h.

**Figure 8 genes-14-01248-f008:**
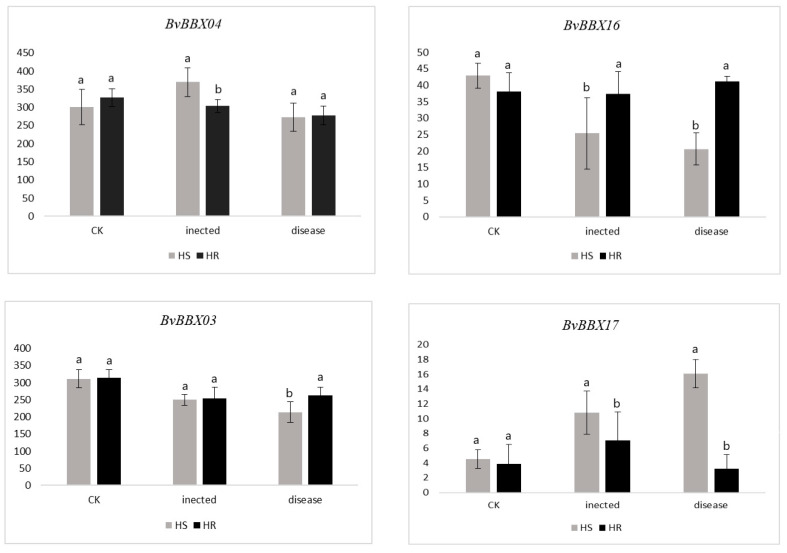
The expression levels of *BvBBX3*, *BvBBX4*, *BvBBX16*, and *BvBBX17* after *C. beticola* infection. Note. “HS” refers to highly susceptible variety; “HR” refers to highly resistant variety; “infected” represents the infection of sugar beet brown spot for 36 h, “disease” represents the infection of sugar beet brown spot for 96 h. The lowercase letters above the bar chart indicate significant differences per data point determined with one-way ANOVA and different letters represent significant differences (*p* < 0.05). Error bars represent standard deviations of the means of triplicates (*n* = 3).

**Table 1 genes-14-01248-t001:** The reaction system of RT-qPCR.

Reagent	Amount
TB Green Premix Ex Taq II (2×)	10 μL
PCR Forward Primer (10 μM)	0.8 μL
PCR Reverse Primer (10 μM)	0.8 μL
ROX Reference Dye (50×)	0.4 μL
DNA Templates	2 μL
Sterile Water	Filling to 20 μL

**Table 2 genes-14-01248-t002:** The specific primers used for RT-qPCR.

Gene	Forward Primer (5′–3′)	Reverse Primer (5′–3′)
*BvBBX01*	AGACGCGCCGAAAAAGAAAA	GGGCGTTCCTATTCCCTCG
*BvBBX02*	AAAAGATTTAGCGTTGACCACC	TGTTGATGGTCTATGTGAGTCCC
*BvBBX03*	CTTCTGCTGCTTGCAAACCT	GATGAGCGCTTACACTGTGAC
*BvBBX04*	CTGATGGTTTGCGGCAAGAG	ACTGACCTGATCTGCACATTT
*BvBBX05*	TGATGGGATGACACGAACGG	TCATCACGCCATGTGGAACA
*BvBBX06*	GCAACGACGTATTGCGAGTC	CCAGATGCAGTCCAAGGTGT
*BvBBX07*	TTTTTCCCGTGCTGTGTGGG	CAAGACAATTGAACTCCGTACCA
*BvBBX08*	TTCCGGATTCGTCTTGATCCC	ATGTTCACCCTTTTGCTTGGT
*BvBBX09*	TTCCACACTGACCATCCCAC	TGCCTTCAGGGATGTGATCG
*BvBBX10*	TTCCACACTGACCATCCCAC	TGCGAGACTGAAGATGTTACCT
*BvBBX11*	CCACCACCGTACCTTTCTCC	TTCCGGCGAAGACGATTCTC
*BvBBX12*	TCAACTTTCCTTTATCTCAACCACA	CGAATTCGTGTTTCCGTCTGG
*BvBBX13*	ACCTCAATGCATCAAATCTTCCA	TCCAGATTTTGCGGGTTCAGT
*BvBBX14*	ACAACAAGGAGATCTTTCGGGG	GGCTCGGTAACCCTGTAAACT
*BvBBX15*	TGTTTCTCGATGCTAATTCGAGG	CGACATGCATGAAAAGCAAACG
*BvBBX16*	TCCCATGGGTAGACCATTTGC	GGCCAACCCAAAACTAACTGC
*BvBBX17*	TCTATGCCTTGCAATAGCCCT	CCCAAGTTCGGGCCTACTTT
*BvActin*	ACTGGTATTGTGCTTGACTC	ATGAGATAATCAGTGAGATC

**Table 3 genes-14-01248-t003:** Physicochemical properties of BBX proteins.

Gene Name	Gene Symbol	Accession Number	Protein Length	pI	MW (Da)	Chr. Location	Exon No.	Subcellular Localization
*BvBBX01*	LOC104888895	XM_010674009.2	328	6.06	35915.31	1	3	Nucleus.
*BvBBX02*	LOC104907374	XM_010696302.2	368	5.41	40858.69	2	2	Nucleus.
*BvBBX03*	LOC104906002	XM_010694679.2	384	6.1	41702.13	2 un	2	Nucleus.
*BvBBX04*	LOC104888829	XM_010673928.2	246	4.94	27226.69	3	3	Nucleus.
*BvBBX05*	LOC104891303	XM_010676978.2	312	5.44	33737.74	4	3	Nucleus.
*BvBBX06*	LOC104907254	XM_010696160.2	201	4.12	22061.55	4 un	1	Nucleus.
*BvBBX07*	LOC104896645	XM_010683418.2	443	6.7	49495.5	6	6	Nucleus.
*BvBBX08*	LOC104897491	XM_010684375.2	517	5.64	57633.24	6	4	Nucleus.
*BvBBX09*	LOC104897745	XM_019250497.1	338	4.58	37833.16	6	4	Nucleus.
*BvBBX10*	LOC104908945	XM_010698113.2	389	5.99	41552.96	7 un	1	Nucleus.
*BvBBX11*	LOC104908648	XM_010697779.2	254	5.92	27896.79	7 un	4	Nucleus.
*BvBBX12*	LOC104902431	XM_010690211.2	379	5.99	41552.96	8	2	Nucleus.
*BvBBX13*	LOC104903341	XM_010691368.2	412	4.97	44928.89	9	6	Nucleus.
*BvBBX14*	LOC104904254	XM_010692457.2	196	5.75	21583.28	9	6	Nucleus.
*BvBBX15*	LOC104904288	XM_010692495.2	262	5.77	29046.26	9	3	Nucleus.
*BvBBX16*	LOC104884672	XM_010669358.2	430	5.64	48855.67	Un	2	Nucleus.
*BvBBX17*	LOC104884884	XM_010669588.2	135	4.33	14743.74	Un	1	Nucleus.

Note. pI, isoelectric point; MW, molecular weight; Chr. Location, chromosome location; Exon No., number of exons; Un, unknown.

**Table 4 genes-14-01248-t004:** The function of the cis-acting elements in the *BvBBXs’* promoter region.

Cis-Acting Element	Number of Genes	Functions of Cis-Acting Element
CAAT-box	17	common cis-acting element in promoter and enhancer regions
TATA-box	17	core promoter element around −30 of transcription start
Box 4	16	part of a conserved DNA module involved in light responsiveness
ABRE	16	cis-acting element involved in abscisic acid responsiveness
G-box	13	cis-acting regulatory element involved in light responsiveness
ARE	13	cis-acting regulatory element essential for anaerobic induction
TCT-motif	11	part of a light-responsive element
CGTCA-motif	10	cis-acting regulatory element involved in MeJA responsiveness
TGACG-motif	10	cis-acting regulatory element involved in MeJA responsiveness
O_2_-site	9	cis-acting regulatory element involved in zein metabolism regulation
TCA-element	9	cis-acting element involved in salicylic acid responsiveness
I-box	8	part of a light-responsive element
GATA-motif	8	part of a light-responsive element
GT1-motif	8	light-responsive element
MBS	7	MYB binding site involved in drought inducibility
TGA-element	6	auxin-responsive element
TC-rich repeats	6	cis-acting element involved in defense and stress responsiveness
CAT-box	5	cis-acting regulatory element related to meristem expression
WUN-motif	5	wound-responsive element
P-box	5	gibberellin-responsive element
AE-box	5	part of a module for light response
CCAAT-box	5	MYBHv1 binding site
LTR	5	cis-acting element involved in low-temperature responsiveness
TCCC-motif	5	part of a light-responsive element
MRE	4	MYB binding site involved in light responsiveness
ACE	4	cis-acting element involved in light responsiveness
ATCT-motif	3	part of a conserved DNA module involved in light responsiveness

## Data Availability

Not applicable.

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
