# Peer review of "Genome-Wide Identification of B-Box Gene Family and Expression Analysis Suggest Its Roles in Responses to Cercospora Leaf Spot in Sugar Beet (Beta Vulgaris L.)"

_genes, 2023, doi:10.3390/genes14061248_

Round 1

Reviewer 1 Report

The manuscript entitled “Genome-wide Identification of B-box Gene Family and Expression Analysis Suggests Their Association with Cercospora leaf spots disease resistance in Sugar beet (Beta Vulgaris L.)by Song et al. aimed to perform a genome-wide identification, evolutionary analysis, sequence characterization, and expression analysis of B-box gene family in sugar beet (Beta Vulgaris L. ). The research work is important, as sugar beet is one of the two major sugar crops in the world. The authors have done a detailed analysis of B-box Gene Family in Sugar beet.

However, I have the following major comments for the authors:

1.      In the Materials and Methods, line 126-129, under the section 2.6. Analysis of Cis-acting Elements of BvBBX Gene Promoter, the authors have mentioned, “The sequence of each sugar beet B-box gene promoter region (2000 bp) was extracted from the sugar beet genome database. The type, number, and function of cis-acting elements were analyzed using the Plant CARE database (https://bioinformatics.psb.ugent.be/webtools/plantcare/) [38].”

Please clearly mention regarding the extraction of 2000 bp upstream promoter sequences.  From which site (point) did you extract the upstream genomic DNA sequences?  The 5’, 3’position, and the nucleotide numbering in the Figure 6 for the promoter cis-element analysis are not correct. Please check it and revise correctly.

The authors have used Plant CARE database for promoter cis-element analysis. However, this database is very old and does not have up to date information. Hence, you are certainly going to miss many important and new cis-elements that are recently identified. The best option is to use the MATCH program in TRANSFAC (geneXplain) which you need to pay for the subscription. However, the authors can also use PlantPAN and PLACE database if you do not have access to TRANSFAC. PlantPAN is quite up to date.

2.      The legend for Figure 6: “Conserved domains of B-box genes family in beet” is not correct.  It is promoter cis-element analysis. Please revise the legend. Please mention about each figure in the text. Some of the figures are missing in the text.

3.      There is no standard deviation in Figure 8. How many samples did you use for this data?

4.      The language needs to be polished. There are many mistakes. Please check the language throughout the manuscript.

For example:

i)                   Line 83: “Download beet genome-wide data, protein sequences, and annotation files from the Ensemble Plants database (https://plants.ensembl.org/index.html) [28].”---please rephrase the sentence.

ii)                 Line 132-134: “When the seedling age was 3 pairs of true leaves, 9 plants of KWS5145 highly susceptible variety and 9 plants of F85621 highly resistant variety with good growth were  selected.”….please rephrase the sentence.

Please refine the language throughout the manuscript. 

Author Response

Dear Editor,

  Thank you for arranging a timely review of our manuscript. We have carefully evaluated the critical comments and thoughtful suggestions. Our manuscript,  Genome-wide Identification of B-box Gene Family and Expression Analysis Suggests Its Roles in Responses to Cercospora leaf spots in Sugar beet (Beta Vulgaris L. ), was checked according to the reviewers' comments, and the itemized response to the reviewer's comments is attached. We use the "Track Changes" so they may be easily identified. Thank you very much for your suggestion.

1. In the Materials and Methods, line 126-129, under the section 2.6. Analysis of Cis-acting Elements of BvBBX Gene Promoter, the authors have mentioned, “The sequence of each sugar beet B-box gene promoter region (2000 bp) was extracted from the sugar beet genome database. The type, number, and function of cis-acting elements were analyzed using the Plant CARE database (https://bioinformatics.psb.ugent.be/webtools/plantcare/) [38].”

Please clearly mention regarding the extraction of 2000 bp upstream promoter sequences.  From which site (point) did you extract the upstream genomic DNA sequences?  The 5’, 3’position, and the nucleotide numbering in the Figure 6 for the promoter cis-element analysis are not correct. Please check it and revise correctly.

The authors have used Plant CARE database for promoter cis-element analysis. However, this database is very old and does not have up to date information. Hence, you are certainly going to miss many important and new cis-elements that are recently identified. The best option is to use the MATCH program in TRANSFAC (geneXplain) which you need to pay for the subscription. However, the authors can also use PlantPAN and PLACE database if you do not have access to TRANSFAC. PlantPAN is quite up to date.

The author's answer:

Thank you very much for you suggestions. The promoter region is 2000bp upstream of the CDS region of the extracted gene. In the article has been modified. In line 391. The 5’, 3’ position and the nucleotide numbering in Figure 6 for the promoter cis-element analysis are not correct. I'm sorry I didn't see the 5‘, 3 position and nucleotide numbering errors.The PlantPAN database seems to be a trans-acting factor, without the function of predicting cis-acting elements.

2. The legend for Figure 6: “Conserved domains of B-box genes family in beet” is not correct.  It is promoter cis-element analysis. Please revise the legend. Please mention about each figure in the text. Some of the figures are missing in the text.

The author's answer:

In line 391-420. The legend in Fig. 6 has been modified. We were really sorry for our careless mistakes. Thank you for your reminder. In the article, we mentioned each cis-acting element in Table 4, which is not detailed enough. Thank you for your suggestion.

3.      There is no standard deviation in Figure 8. How many samples did you use for this data?

The author's'answer:

In line 465-466. The standard deviation was supplemented in the picture, and each treatment was repeated three times in the experiment.

4. The language needs to be polished. There are many mistakes. Please check the language throughout the manuscript.

For example:

i)Line 83: “Download beet genome-wide data, protein sequences, and annotation files from the Ensemble Plants database (https://plants.ensembl.org/index.html) [28].”---please rephrase the sentence.

ii)Line 132-134: “When the seedling age was 3 pairs of true leaves, 9 plants of KWS5145 highly susceptible variety and 9 plants of F85621 highly resistant variety with good growth were  selected.”….please rephrase the sentence.

The author's answer:

Thanks for your careful checks. We are sorry for our carelessness. The sentence has been modified.

Line123-125. Download sugar beet DNA sequences, protein sequences, and annotation files from the Ensembl Plants database (https://plants.ensembl.org/index.html) [28].

Line 202-204. When the seedling age was 3 pairs of true leaves, strains of KWS5145(highly susceptible to Cercospora leaf spots)and 9 strains of F85621(highly resistant to Cercospora leaf spots)were selected respectively.

  We tried our best to improve the manuscript and made some changes to the manuscript. These changes will not influence the content and framework of the paper. And here we did not list the changes but marked them in red in the revised paper. We appreciate for reviewers' warm work earnestly and hope that the correction will meet with approval.

  We hope that the revised manuscript has addressed all the criticisms raised by the editor and the reviewers and that the manuscript is now suitable for publication in Genes.

Yours sincerely,

He Song

12 May. 2023

Reviewer 2 Report

A manuscript entitled “Genome-wide Identification of B-box Gene Family and Expression Analysis Suggests Their Association with Cercospora Leaf spots disease resistance in Sugar beet (Beta Vulgaris L. )” was reviewed for publication in Genes.

 The authors detected 17 genes that have homology to the Arabidopsis B-box family in the sugar beet genome. The putative B-box proteins were characterized in physicochemical analysis, domain analysis, and phylogenetic analysis. The gene expression analysis was performed with qPCR together with cis-element analysis indicating some candidate genes for bacterial resistance.

 Questions and Comments

1.     The title “Their Association with Cercospora leaf spots disease resistance” sounds like a too strong relationship between disease resistance and gene expression. Several papers describing gene expression’s relationship to phenotypes use a more objective expression such as “Its Roles in Responses to~” or “their expression profiles under XXXX. I would suggest the expression in the title to be changed.

2.     To improve quality, English editing is recommended.

For example,

L83: Download beet genome-wide data, protein sequences, and annotation files from the Ensembl Plants database (https://plants.ensembl.org/index.html)

L138: Using spray inoculation.

L167: Save in an environment of -80 C. Unified RT-166 qPCR.-

L248: To further study the conservation and diversity of sugar beet BBX protein structure.

L.356: Although they all lack a B-box do-356 main[5].

----please try to make a complete sentence.

L86: Please describe the URL of the Pfam database. There is a notation that the Pfam database is now provided through InterPro at EBI(https://www.ebi.ac.uk/interpro/).

L90: Is there a coverage cut-off? If so, please indicate it. Or did you choose the top hit sequence?

L91: e=1xe-3 should be corrected as “the minimum E value was e-3 ”

L132 & L144: Please describe the experimental background more in detail.  On KWS5145 (highly susceptible variety) and F85621 (highly resistant variety), do you have any information(reference) on susceptibility phenotypes for both varieties? What do 36hr and 96hr after inoculation mean for sugar beet plants, such as bacterial propagation or the emergence of disease symptoms?

L147~L167: Although the numbering of each sample used in the experiment is described, it did not appear in the Results section because I imagine the data from those three samples were averaged. Therefore, it is enough to describe here as “Results of three independent experiments were averaged”.

L305: Where is the description explaining Fig.8?  Since Fig.8 shows the expression levels of BvBBX3, BvBBX4, BvBBX16, and BvBBX17, there should be a remark “Fig. 8” here.

L167: About the specific primers used for qPCR, did you confirm primer specificity using in silico search or single band identification in Reverse-transcription PCR using cDNA?

L191-202: After “CONSTANS--- should be moved to the Discussion section, since it is not the direct results.

L208: molecular length-----number of amino acids.

L209: Please describe the software for predicting subcellular location e.g.TargetP.

L221: Please indicate the name of each chromosome in Fig.1

  “four members on chromosomes 6 and 9.” Which four BBX genes were found in Chr.6 and 9?  Both chromosomes in which multiple BBX genes are mapped seem to have three BBXs.

L242 & L299:Figs 3 and 6, beet----sugar beet.

Fig.3 ----Please indicate which figure shows which result. 

L249: MEME online software Please refer to URL.

L323: “Therefore, the differential expression of BvBBX4, BvBBX3, BvBBX16, and BvBBX17 may make sugar beet produce disease resistance”.

Since this is a part of the Results Section, I think the result that indicates there is a co-expression relationship between the expression of some BBX genes and disease resistance does not directly leads to a cause-relationship unless other indirect evidence will be showed. A more unbiased description will be suitable here. 

To improve quality, English editing is recommended.

For example,

L83: Download beet genome-wide data, protein sequences, and annotation files from the Ensembl Plants database ( https://plants.ensembl.org/index.html)

L138: Using spray inoculation.

L167: Save in an environment of -80 C. Unified RT-166 qPCR.-

L248: To further study the conservation and diversity of sugar beet BBX protein structure.

L.356: Although they all lack a B-box do-356 main[5].

----please try to make a complete sentence.

L208: molecular length-----number of amino acids.

 L242 & L299:Figs 3 and 6, beet----sugar beet.

Author Response

Dear Editor,

  We feel graet thinks for your professional review work on our articies. As you are concerned, there are several problem that need to be addessed. According to your nice suggestions, we have made extensive corrections to our previous draft. The detailed corrections are listed below. We use the "Track Changes" so they may be easily identified. Thank you very much for your suggestion.

1. The title “Their Association with Cercospora leaf spots disease resistance” sounds like a too strong relationship between disease resistance and gene expression. Several papers describing gene expression’s relationship to phenotypes use a more objective expression such as “Its Roles in Responses to~” or “their expression profiles under XXXX. I would suggest the expression in the title to be changed.

Answer: We think this is an excellect suggestion. The title has been modified to Genome-wide Identification of B-box Gene Family and Expression Analysis Suggests Its Roles in Responses to Cercospora leaf spots in Sugar beet (Beta Vulgaris L. ).

2. To improve quality, English editing is recommended.

For example,

L83: Download beet genome-wide data, protein sequences, and annotation files from the Ensembl Plants database (https://plants.ensembl.org/index.html)

L138: Using spray inoculation.

L167: Save in an environment of -80 C. Unified RT-166 qPCR.-

L248: To further study the conservation and diversity of sugar beet BBX protein structure.

L.356: Although they all lack a B-box do-356 main[5].

----please try to make a complete sentence.

Answer:Thank you very much for watching carefully. Some sentences may not be complete enough. We have revised him completely. If there are other problems, please criticize and correct them.

L86: Please describe the URL of the Pfam database. There is a notation that the Pfam database is now provided through InterPro at EBI(https://www.ebi.ac.uk/interpro/).

Answer:L128:We add the URL of the Pfam database to the article.

L90: Is there a coverage cut-off? If so, please indicate it. Or did you choose the top hit sequence?

Answer:L134-135:The results of 31 protein sequences in the results are the addition of the sequences obtained by the two methods, which have been explained in the paper.

L91: e=1xe-3 should be corrected as “the minimum E value was e-3 ”

Answer: L134: We feel sorry for our carelessness. In our resubmitted manuscript, the type is revised. Thanks for your correction. 

L132 & L144: Please describe the experimental background more in detail.  On KWS5145 (highly susceptible variety) and F85621 (highly resistant variety), do you have any information(reference) on susceptibility phenotypes for both varieties? What do 36hr and 96hr after inoculation mean for sugar beet plants, such as bacterial propagation or the emergence of disease symptoms?

Answer: L202-204&226&235: The information of KWS5145 and F85621 has been changed in the text, and the information on susceptibility phenotypes has been supplemented.36h means Cercospora beticola began to colonize. 96h means the symptoms began to appear. In the article has carried on the supplementary explanation.

L147~L167: Although the numbering of each sample used in the experiment is described, it did not appear in the Results section because I imagine the data from those three samples were averaged. Therefore, it is enough to describe here as “Results of three independent experiments were averaged”.

Answer: Indeed, each treatment had three replicates, and the results were described by the mean of three replicates.

L305: Where is the description explaining Fig.8?  Since Fig.8 shows the expression levels of BvBBX3, BvBBX4, BvBBX16, and BvBBX17, there should be a remark “Fig. 8” here.

Answer:L457-465:A description of Figure 8 is added.

L167: About the specific primers used for qPCR, did you confirm primer specificity using in silico search or single band identification in Reverse-transcription PCR using cDNA?

Answer:After reverse transcription PCR using cDNA, the electrophoresis result was a band, which proved the specificity of the primers.

L191-202: After “CONSTANS--- should be moved to the Discussion section, since it is not the direct results.

Answer:L486-494:We have changed the content after “CONSTANT” to the discussion area.

L208: molecular length-----number of amino acids.

Answer:L289: We sincerely thank the reviewer for careful reading. As suggested by the reviewer, we have corrected the "molecular length" into"number of amino acids" 

L209: Please describe the software for predicting subcellular location e.g.TargetP.

Answer: L146-L148: Predicting geological location software is Cell-PLoc 2.0, described in 2.1 where materials and methods are located.

L221: Please indicate the name of each chromosome in Fig.1

  “four members on chromosomes 6 and 9.” Which four BBX genes were found in Chr.6 and 9?  Both chromosomes in which multiple BBX genes are mapped seem to have three BBXs.

Answer:The names of the chromosomes in Figure 1 are all identified. There are three BvBBXs on chromosomes 6 and 9, and I am sorry for our carelessness. The BvBBXs on each chromosome in the article were supplemented.

L242 & L299:Figs 3 and 6, beet----sugar beet.

Answer: We sincerely thank the reviewer for careful reading. As suggested by the reviewer, we have corrected the "beet" into"sugar beet" 

Fig.3 ----Please indicate which figure shows which result. 

Answer:L331-332:  The article has indicated which figure shows which result.

L249: MEME online software Please refer to URL.

Answer:L239-240: The URL of MEME has been supplemented in the article.(https://meme-suite.org/meme/doc/meme.html)

L323: “Therefore, the differential expression of BvBBX4, BvBBX3, BvBBX16, and BvBBX17 may make sugar beet produce disease resistance”.

Since this is a part of the Results Section, I think the result that indicates there is a co-expression relationship between the expression of some BBX genes and disease resistance does not directly leads to a cause-relationship unless other indirect evidence will be showed. A more unbiased description will be suitable here. 

Answer: L463-465: Indeed, the results can not be seen that increased gene expression leads to disease resistance, can only prove that the two are related but no causal relationship, has been changed in the text.

  We tried our best to improve the manuscript and made some changes to the manuscript. These changes will not influence the content and framework of the paper. And here we did not list the changes but marked them in red in the revised paper. We appreciate for reviewers' warm work earnestly and hope that the correction will meet with approval.

  We hope that the revised manuscript has addressed all the criticisms raised by the editor and the reviewers and that the manuscript is now suitable for publication in Genes.

Yours sincerely,

He Song

12 May. 2023

Round 2

Reviewer 1 Report

Thank you author for trying to revise the manuscript.

However, I have the following major comments for the authors:

1.     The authors have responded that ‘The promoter region is 2000bp upstream of the CDS region of the extracted gene.’

Actually, the promoter sequence is always in the upstream of the Transcription Start Site (TSS). However, the authors have taken 2000 bp upstream of CDS (ATG site), which is Translation initiation site (TIS). Hence, the sequence they have taken could be a part of Exon 1 and a part of promoter or only Exon 1 depending on the gene length. Hence, the cis-element detected in those regions are not correct, as the sequence is not right. Sometime, some distal cis-elements are present in the downstream of promoter sequence, but majority are present in the upstream part of promoter sequence. Hence, please extract the promoter sequence from the upstream of TSS and perform the cis-element enrichment analysis and revise the relevant information. Accordingly, you need to revise the results and discussion part that is based on your new promoter sequence analysis and experiments.

2.     The authors have used Plant CARE database for promoter cis-element analysis. However, this database is very old and does not have up to date information. Hence, you are certainly going to miss many important and new cis-elements that are recently identified. The best option is to use the MATCH program in TRANSFAC (geneXplain) which you need to pay for the subscription. However, the authors can also use PlantPAN and PLACE database if you do not have access to TRANSFAC. PlantPAN is quite up to date.

The authors have mentioned that: “The PlantPAN database seems to be a trans-acting factor, without the function of predicting cis-acting elements”….it is not true. You can do promoter analysis for cis-element identification using PlantPAN database.

3.     The 5’, 3’ position and the nucleotide numbering in Figure 6 for the promoter cis-element analysis are not correct. Please revise it correctly.

4.     Please check the language throughout the manuscript.

Need to check and refine the language throughout the manuscript.

Author Response

Dear Editor,

We feel great thinks for your professional review work on our articles. As you are concerned, there are several problem that need to be addressed. According to your nice suggestions, we have made extensive corrections to our previous draft. The detailed corrections are listed below. We use the "Track Changes" so they may be easily identified. Thank you very much for your suggestion.

  1. The authors have responded that ‘The promoter region is 2000bp upstream of the CDS region of the extracted gene.’

Actually, the promoter sequence is always in the upstream of the Transcription Start Site (TSS). However, the authors have taken 2000 bp upstream of CDS (ATG site), which is Translation initiation site (TIS). Hence, the sequence they have taken could be a part of Exon 1 and a part of promoter or only Exon 1 depending on the gene length. Hence, the cis-element detected in those regions are not correct, as the sequence is not right. Sometime, some distal cis-elements are present in the downstream of promoter sequence, but majority are present in the upstream part of promoter sequence. Hence, please extract the promoter sequence from the upstream of TSS and perform the cis-element enrichment analysis and revise the relevant information. Accordingly, you need to revise the results and discussion part that is based on your new promoter sequence analysis and experiments.

Answer: The cis-acting element, based on its definition, is not necessarily a promoter region, but can also be in introns, or in adjacent genes. So it seems not directly related to the promoter. However, the promoter is by definition near the region where RNA polymerase ( such as pol II ) is recruited and bound. This region should have more transcription factors ( trans-acting factors ) and transcription regulators, so naturally there are more cis-acting elements. Regarding the selection of the promoter region, it is true that the promoter sequence is always upstream of the Transcription Start Site ( TSS ). However, different transcripts of the same gene have different Transcription Start Sites, so it is difficult to find the real promoter on TSS. Therefore, 2000 bp upstream of the CDS was taken as the promoter, which may include some UTR regions. But there seems to be no better way. In many articles, the analysis of cis-acting elements takes 1kb-2kb or longer upstream of the translation initiation codon ( ATG ). For example, Genome-Wide Characterization and Anthocyanin-Related Expression Analysis of the B-BOX Gene Family in Capsicum annuum L.https://www.frontiersin.org/articles/10.3389/fgene.2022.847328/full, The Function of BBX Gene Family under Multiple Stresses in Nicotiana tabacum. https://www.mdpi.com/2073-4425/13/10/1841. Therefore, I think it is possible to take 2000bp upstream of CDS for the analysis of cis-acting elements.

  1. The authors have used Plant CARE database for promoter cis-element analysis. However, this database is very old and does not have up to date information. Hence, you are certainly going to miss many important and new cis-elements that are recently identified. The best option is to use the MATCH program in TRANSFAC (geneXplain) which you need to pay for the subscription. However, the authors can also use PlantPAN and PLACE database if you do not have access to TRANSFAC. PlantPAN is quite up to date.

The authors have mentioned that: “The PlantPAN database seems to be a trans-acting factor, without the function of predicting cis-acting elements”….it is not true. You can do promoter analysis for cis-element identification using PlantPAN database.

Answer: We agree with you that the PlantPAN has many new cis-acting elements ,however the elements searched using plantPAN may be difficult to match with the cis-acting elements in PlantCARE. It seems to be the name of the element named by the database itself, and there is no cis-acting element function, only the information of which transcription factor can be combined. I cannot filter the components I am interested in and show them in the figure. In this article is not good for subsequent analysis. The analysis of cis-acting elements in PlantCARE has been very mature. Most people use PlantCARE for analysis. For example, in many articles on the identification and analysis of B-box gene families, PlantCARE database is used. Thank you very much for letting me know a new database for analyzing cis-elements.

  1. The 5’, 3’ position and the nucleotide numbering in Figure 6 for the promoter cis-element analysis are not correct. Please revise it correctly.

Answer: L326. We feel sorry for our carelessness. In our resubmitted manuscript, the figure 6 is revised. Thanks for your correction.

  1. Please check the language throughout the manuscript.

Answer: Thanks for your suggestion. We have tried our best to polish the language in the revised manuscript.

We tried our best to improve the manuscript and made some changes to the manuscript. These changes will not influence the content and framework of the paper. And here we did not list the changes but marked them in red in the revised paper. We appreciate for reviewers' warm work earnestly and hope that the correction will meet with approval.

We hope that the revised manuscript has addressed all the criticisms raised by the editor and the reviewers and that the manuscript is now suitable for publication in Genes.

Yours sincerely,

He Song

26 May. 2023

Reviewer 2 Report

The revised manuscript was reviewed.

With the extensive editing, the manuscript has been much improved.

However, I still found some non-standard sentences and those have not been changed so far.

 1.    The italic description should be used if you use a plant name.

Beta Vulgaris and Arabidopsis thaliana should be described in italics.

 2.    Please do not use imperative sentences, since this is not the manual for the experiments.

L83(NewL123): Download sugar beet DNA sequences-----

We downloaded” or “Sugar beet sequences----were downloaded.” are appropriate.

L138(NewL210): Using spray method. “The fungi were infected using spray method” should be used.

L167(New L166): Save in an environment of -80 C.

---------"All the samples were saved at -80 C.”

(NewL130): Obtain the most-----,” and obtained the most-----

 3.    L248(New L349): To deepen our understanding of the diversity and conservatism of sugar beet BBX protein structure. The MEME online program (https://meme suite.org/meme/doc/meme.html) was used to predict the conserved motifs of the family proteins.

-----Please check the use of the (, comma) and (. period) in your word processor. Here, “protein structure, the MEME online---” should be used.

4, L147~L167:

Original comment:

Although the numbering of each sample used in the experiment is described, it did not appear in the Results section because I imagine the data from those three samples were averaged. Therefore, it is enough to describe here as “Results of three independent experiments were averaged”.

Answer:

Indeed, each treatment had three replicates, and the results were described by the mean of three replicates.

-------My suggestion is that this part could be trimmed by not describing the sample name, such as: (the example is only for the control group) ”The control group (no Cercospora infection):1g of the leaves in the same growth status were individually excised from three plants(that is, 3 samples in each group) for each sugar beet variety. Each sample was stored in a separate centrifuge tube at -80 °C.”

 Please consider.

 5.    I cannot see the word Figure 4, 6, and Figure 8 in the main text. Please insert where appropriate.

Although most sentences are improved, there left some expressions, which I showed in the comments, are not for standard scientific publication.

Author Response

Dear Editor,

  Thank you for arranging a timely review of our manuscript. We have carefully evaluated the critical comments and thoughtful suggestions. Our manuscript, Genome-wide Identification of B-box Gene Family and Expression Analysis Suggests Its Roles in Responses to Cercospora leaf spots in Sugar beet (Beta Vulgaris L.), was checked according to the reviewers' comments, and the itemized response to the reviewer's comments is attached. We use the "Track Changes" so they may be easily identified. Thank you very much for your suggestion.

1. The italic description should be used if you use a plant name.

Beta Vulgaris and Arabidopsis thaliana should be described in italics.

Answer: We sincerely thank the reviewer for careful reading. As suggested by the reviewer, We have described Arabidopsis thaliana and Beta Vulgaris using italics.

2. Please do not use imperative sentences, since this is not the manual for the experiments.

L83(NewL123): Download sugar beet DNA sequences-----

“We downloaded” or “Sugar beet sequences----were downloaded.” are appropriate.

Answer: L81. Thank you for your suggestion. We have modified the sentence to “We downloaded”.

L138(NewL210): Using spray method. “The fungi were infected using spray method” should be used.

Answer: L140. We sincerely thank the reviewer for careful reading. We have corrected the “Using spray method” into “The fungi were infected using spray method”.

L167(New L166): Save in an environment of -80 C.-------"All the samples were saved at -80 C.”

Answer: L172. We have corrected the “Save in an environment of -80 C” into “All the samples were saved at -80 C.”.

(NewL130): Obtain the most-----,” and obtained the most-----

Answer: L86. We have corrected the “Obtain the most-----,” into “and obtained the most-----”.

3. L248(New L349): To deepen our understanding of the diversity and conservatism of sugar beet BBX protein structure. The MEME online program (https://meme suite.org/meme/doc/meme.html) was used to predict the conserved motifs of the family proteins.

-----Please check the use of the (, comma) and (. period) in your word processor. Here, “protein structure, the MEME online---” should be used.

Answer: L271. Furthermore, by following the reviewer’s suggestion, we modify the sentence to:“protein structure, the MEME online---”

4. L147~L167:

Original comment:

Although the numbering of each sample used in the experiment is described, it did not appear in the Results section because I imagine the data from those three samples were averaged. Therefore, it is enough to describe here as “Results of three independent experiments were averaged”.

Answer:

Indeed, each treatment had three replicates, and the results were described by the mean of three replicates.

-------My suggestion is that this part could be trimmed by not describing the sample name, such as: (the example is only for the control group) ”The control group (no Cercospora infection):1g of the leaves in the same growth status were individually excised from three plants(that is, 3 samples in each group) for each sugar beet variety. Each sample was stored in a separate centrifuge tube at -80 °C.”

 Please consider.

Answer: L157~L190. We are grateful for this suggestion. Indeed, this part describes the repeated sample number is not clear enough. This position has been described in a more concise language, deleting duplicate sample names. The descriptions of the control group, The method descriptions of the early infected test group and the disease test group were modified.

5. I cannot see the word Figure 4, 6, and Figure 8 in the main text. Please insert where appropriate.

Answer: L267, L275, L310, L350. We were really sorry for our careless mistakes. Thank you for your reminder. We have marked the word Figure 4, 6, and Figure 8 in the right place in the main text.

  We tried our best to improve the manuscript and made some changes to the manuscript. These changes will not influence the content and framework of the paper. And here we did not list the changes but marked them in red in the revised paper. We appreciate for reviewers' warm work earnestly and hope that the correction will meet with approval.

  We hope that the revised manuscript has addressed all the criticisms raised by the editor and the reviewers and that the manuscript is now suitable for publication in Genes.

Yours sincerely,

He Song

26 May. 2023
